# Gα_s_-Coupled CGRP Receptor Signaling Axis from the Trigeminal Ganglion Neuron to Odontoblast Negatively Regulates Dentin Mineralization

**DOI:** 10.3390/biom12121747

**Published:** 2022-11-24

**Authors:** Natsuki Saito, Maki Kimura, Takehito Ouchi, Tatsuya Ichinohe, Yoshiyuki Shibukawa

**Affiliations:** 1Department of Physiology, Tokyo Dental College, 2-9-18, Kanda-Misaki-cho, Chiyoda-ku, Tokyo 101-0061, Japan; 2Department of Dental Anesthesiology, Tokyo Dental College, 2-9-18, Kanda-Misaki-cho, Chiyoda-ku, Tokyo 101-0061, Japan

**Keywords:** axon reflex, cAMP, CGRP, dental pulp, inflammatory response, odontoblasts, trigeminal ganglion

## Abstract

An inflammatory response following dental pulp injury and/or infection often leads to neurogenic inflammation via the axon reflex. However, the detailed mechanism underlying the occurrence of the axon reflex in the dental pulp remains unclear. We sought to examine the intracellular cyclic adenosine monophosphate (cAMP) signaling pathway in odontoblasts via the activation of G_s_ protein-coupled receptors and intercellular trigeminal ganglion (TG) neuron–odontoblast communication following direct mechanical stimulation of TG neurons. Odontoblasts express heterotrimeric G-protein α-subunit Gα_s_ and calcitonin receptor-like receptors. The application of an adenylyl cyclase (AC) activator and a calcitonin gene-related peptide (CGRP) receptor agonist increased the intracellular cAMP levels ([cAMP]_i_) in odontoblasts, which were significantly inhibited by the selective CGRP receptor antagonist and AC inhibitor. Mechanical stimulation of the small-sized CGRP-positive but neurofilament heavy chain-negative TG neurons increased [cAMP]_i_ in odontoblasts localized near the stimulated neuron. This increase was inhibited by the CGRP receptor antagonist. In the mineralization assay, CGRP impaired the mineralization ability of the odontoblasts, which was reversed by treatment with a CGRP receptor antagonist and AC inhibitor. CGRP establishes an axon reflex in the dental pulp via intercellular communication between TG neurons and odontoblasts. Overall, CGRP and cAMP signaling negatively regulate dentinogenesis as defensive mechanisms.

## 1. Introduction

The G-protein-coupled receptor (GPCR)-regulated adenylyl cyclase (AC) signal transduction pathways induce intracellular cyclic adenosine monophosphate (cAMP) signaling in various cells [1,2,3]. G-proteins are composed of α, β, and γ subunits, which are mainly divided into Gα_s_, Gα_i_, Gα_q_, and Gα_12/13_. The many conformations of the receptor lead to a variety of highly specialized downstream signaling cascades [4]. GPCRs induce two main signaling pathways: cAMP and phosphatidylinositol signaling pathways. Gα_s_ and Gα_i_ regulate the cAMP-generating enzyme AC. In fact, Gα_s_ activates AC, while Gα_i_ inhibits AC, leading to increases or decreases in intracellular cAMP ([cAMP]_i_) levels. The diffusible intracellular second messenger system stimulates or inhibits downstream signaling, thereby enabling further biological effects [4].

Calcitonin gene-related peptide (CGRP), which is expressed in various organs such as the brain, heart, liver, and spleen as well as skeletal muscle and dental pulp tissue, controls general circulation by regulating blood flow as a vasodilator [5]. CGRP has a close reciprocal interaction with the sympathetic nervous system in the periphery [5,6,7]. CGRP receptors comprise the calcitonin receptor-like receptor (CALCRL) and receptor activity modifying protein 1 (RAMP1). Notably, CALCRL expression is necessary for the full functionality of the CGRP receptor [5,8]. CGRP receptor activation increases blood flow and enhances vasodilation by producing nitric oxide via the Gα_s_ pathway [5].

Dental pulp is soft tissue of cranial neural crest-derived mesenchymal origin, enclosed by rigid mineralized dentin, ultimately residing in a low-compliance environment. Once an inflammatory response occurs in the dental pulp, the internal tissue pressure increases, which mechanically stimulates the dental pulp cells, including nerve terminals and odontoblasts [7]. The neuropeptide CGRP, which localizes in the terminal of unmyelinated C neuron, is released owing to the inflammation response from sensitized nerve terminals due to the antidromic conduction of an action potential to the nerve endings, known as the “axon reflex”, to increase and prolong the ongoing inflammatory response [7]. The antidromic release of CGRP via axon reflexes induces neurogenic vasodilation in pulpal tissue, potentiating the inflammatory response [9]. Thus, the inflammatory response in dental pulp (pulpitis) might be due to neurogenic inflammation and the axon reflex. Therefore, CGRP is one of the principal factors in axon reflex during pulpitis [10,11].

Odontoblasts differentiate from dental pulp stem cells and localize to the outermost layer of the dental pulp. Odontoblasts drive not only physiological and developmental but also pathological tertiary dentin formation as biological defensive dentin formation. Odontoblasts play an important role in the sensitivity of tooth pain (dentinal sensitivity) through signal communication between odontoblasts and trigeminal ganglion (TG) neurons as sensory receptor cells [12,13]. Although previous studies suggest that CGRP and its receptor activation play roles in the development of an axon reflex that leads to neurogenic inflammation of the dental pulp [7,9], details regarding CGRP and its receptor function in dental pulp cells (i.e., TG neurons and odontoblasts), which determine whether dentin regeneration occurs in response to pulpitis, remain unclear. Further, direct evidence of the occurrence of axon reflexes in the dental pulp via neuropeptide signaling is still lacking.

The present study aimed to determine the role of the CGRP–CGRP receptor axis in axon reflex through measurements of [cAMP]_i_ levels via Gα_s_ protein-coupled CGRP receptor activation in odontoblasts. To model and mimic the influence of tissue pressure increase via an inflammatory response in dental pulp, direct mechanical stimulation was applied to TG neurons, and the [cAMP]_i_ response from the odontoblasts approximating the stimulated TG neurons was measured using an odontoblast–neuron co-culture system.

## 2. Materials and Methods

### 2.1. Ethical Approval

This study was approved by the Animal Research Ethics Committee and the Committee for Recombinant DNA Research of Tokyo Dental College (numbers 200301, 210301, and DNA1805). All animals were treated in accordance with the Guiding Principles for the Care and Use of Animals in the field of physiological sciences, which was approved by the Council of the Physiological Society of Japan and the American Physiological Society.

### 2.2. Solutions and Reagents

A solution containing 136 mM NaCl, 5 mM KCl, 2.5 mM CaCl_2_, 0.5 mM MgCl_2_, 10 mM 2-[4-(2-Hydroxyethyl)-1-piperazinyl] ethanesulfonic acid (HEPES), 10 mM glucose, and 12 mM NaHCO_3_ (pH = 7.4 with tris [hydroxymethyl] aminomethane) was used as the standard extracellular solution (ECS). A high-K^+^ solution (91 mM NaCl, 50 mM KCl, 2.5 mM CaCl_2_, 0.5 mM MgCl_2_, 10 mM HEPES, 10 mM glucose, and 12 mM NaHCO_3_; pH = 7.4) was used to discern TG neurons from glial cells by the activation of depolarization-induced increases in the concentration of intracellular free Ca^2+^ in the neurons. An AC activator, forskolin (FSK) [14,15]; an AC inhibitor, SQ22536 [16,17]; a non-selective CGRP receptor agonist, CGRP (rat) [18]; and a non-selective CGRP receptor antagonist, BIBN 4096 [19], were obtained from R&D Systems, Inc. (Minneapolis, MN, USA). Stock solutions of FSK, SQ22536, and BIBN 4096 were prepared in dimethyl sulfoxide using ultrapure water (Millipore, MA, USA). Stock solutions were diluted with the standard ECS to an appropriate concentration before use. ECS or ECS-containing drugs were administered using a gravity-fed perfusion system (Warner Instruments, Holliston, MA, USA). All other reagents were purchased from Sigma Chemical Co., (St. Louis, MO, USA), unless otherwise indicated.

### 2.3. Dental Pulp Slice Preparation

Dental pulp slice preparations were obtained from newborn Wistar rats (aged 4–8 days) [12,20,21,22]. Decapitation was performed under isoflurane (3%) and pentobarbital sodium anesthesia (25 mg/kg, administered via intraperitoneal injection). To reduce the number of rats, both male and female rats were used in this study. The mandibles were dissected, and the hemimandibles, embedded in alginate impression material, were sectioned transversely through the incisor at a thickness of 500 μm using a standard vibration tissue slicer (ZERO-1; Dosaka EM, Kyoto, Japan). The mandibular sections were sliced to ensure direct visibility of the dentin and enamel between the bone tissue and dental pulp. We selected mandible sections with thin dentin (but with enamel and dentin distinguishable under a microscope) to avoid cellular damage in odontoblasts. The surrounding impression material, bone tissue, enamel, and dentin were carefully removed, and the remaining dental pulp slices were obtained. Pulp slices were treated with ECS containing 0.17% collagenase and 0.03% trypsin at 37 °C for 30 min. To measure [cAMP]_i_, enzymatically treated dental pulp slices were plated in culture dishes; cultured in α-MEM containing 10% fetal bovine serum (FBS), 5% horse serum, 1% amphotericin B, and 1% penicillin-streptomycin (Life Technologies Co., Grand Island, NY, USA); and maintained at 37 °C in a 5% CO_2_ incubator for 24–40 h.

### 2.4. Isolation of the TG Neurons

TG neurons were isolated from neonatal Wistar rats (aged 4–7 days) of both sexes under isoflurane (3%) and pentobarbital sodium anesthesia (50 mg/kg, intraperitoneal injection) [12,23,24,25,26]. The isolated cells were dissociated by enzymatic treatment with Hank’s balanced salt solution (HBSS) (137 mM NaCl, 5.0 mM KCl, 2.0 mM CaCl_2_, 0.5 mM MgCl_2_, 0.44 mM KH_2_PO_4_, 0.34 mM Na_2_HPO_4_, 4.17 mM NaHCO_3_, and 5.55 mM glucose; pH was adjusted to 7.4 using tris) containing 20 U/mL papain (Worthington, Lakewood, NJ, USA) for 20 min at 37 °C, followed by dissociation by trituration. Primary cultures of TG cells were performed using Leibovitz’s L-15 medium (Life Technologies Co.) containing 10% FBS, 1% amphotericin B, 1% penicillin-streptomycin, 24 mM NaHCO_3_, and 30 mM glucose (pH 7.4). TG cells were incubated and maintained for 48 h at 37 °C in a humidified atmosphere containing 95% air and 5% CO_2_.

### 2.5. Immunofluorescence Analysis

For immunofluorescence, isolated rat odontoblasts and TG cells were cultured in 8-well glass chambers (AGC, Shizuoka, Japan) and maintained under culture conditions of 37 °C and 5% CO_2_ for 48 h. Immunofluorescence staining was conducted after mechanical stimulation (described in the section below). The cells were fixed with 4% paraformaldehyde (FUJIFILM Wako Pure Chemical Co., Osaka, Japan) and washed with phosphate-buffered saline (PBS) (Life Technologies Co.). After 10–15 min of incubation with 0.1–0.3% Triton X-100 (Sigma Aldrich) and a blocking reagent (Nacalai Tesque, Kyoto, Japan) at room temperature, the following primary antibodies were added and incubated for 3–4 h at room temperature or overnight at 4 °C: rabbit monoclonal anti-CALCRL (Life Technologies Co.; 703811, 8H9L8, 1:200), mouse monoclonal anti-CGRP (Santa Cruz Biotechnology, Inc., Santa Cruz, CA, USA; sc-57053, 4901, 1:200), mouse monoclonal anti-dentin sialophosphoprotein (DSPP) (Santa Cruz Biotechnology, Inc.; sc-73632, LFMb-21, 1:200), rabbit polyclonal anti-DSPP (Bioss, Woburn, MA, USA; bs-8557R, 1:200), and rabbit polyclonal anti-heterotrimeric G-protein α-subunit Gα_s_ (Gnas) (ABclonal, Woburn, MA, USA; A5546, 1:200). Immunofluorescence analysis was performed for the co-culture condition in a 35 mm adherent culture system (Ibidi, Fitchburg, WI, USA) after measurement of the [cAMP]_i_ and intracellular Ca^2+^ concentration ([Ca^2+^]_i_) in odontoblasts and TG neurons, cell fixation, and treatment. Mouse monoclonal anti-CGRP and rabbit polyclonal anti-neurofilament heavy chain (NF-H) antibodies (Merck Millipore, Darmstadt, Germany; AB1989, 1:100) were used for staining. The secondary antibodies used for immunofluorescence analysis were Alexa Fluor^®^ 488 donkey anti-mouse (Life Technologies Co.; #A21202), Alexa Fluor^®^ 555 donkey anti-mouse (Life Technologies Co.; #A31570), Alexa Fluor^®^ 568 donkey anti-mouse (Life Technologies Co.; #A10037), Alexa Fluor^®^ 488 donkey anti-rabbit (Life Technologies Co.; #A21206), Alexa Fluor^®^ 555 donkey anti-rabbit (Life Technologies Co.; #A31572), Alexa Fluor^®^ 568 donkey anti-rabbit (Life Technologies Co.; #A10042), and Alexa Fluor^®^ 647 donkey anti-rabbit (Life Technologies Co.; #A31573). The secondary antibodies were added and incubated for one hour at room temperature. The stained samples were mounted in mounting reagent with 4,6-diamidino-2-phenylindole (DAPI) (Abcam, Cambridge, UK; ab104139). Immunostaining was analyzed using a fluorescence microscope (Keyence, Osaka, Japan; X710).

### 2.6. Preparation of the Odontoblast–TG Neuron Co-Culture

Primary cultured dental pulp slices, including odontoblasts, were incubated for 24 h at 37 °C and 5% CO_2_ in alpha-minimum essential medium (α-MEM) containing an mNeon-based cAMP sensor, cADDis (Montana Molecular, Bozeman, MT, USA), with 5 mM sodium butyrate (Montana Molecular). Thereafter, the slices were rinsed with fresh ECS. Primary cultured TG neurons were loaded with HBSS containing 10 μM fura-2-acetoxymethyl ester (fura-2-AM; Dojindo Laboratories, Kumamoto, Japan) and 0.1% (*w*/*v*) Pluronic Acid F-127 (F-127; Life Technologies Co.) for 60 min. TG neurons bathed in fura-2-AM/F-127 containing HBSS were resuspended and subsequently rinsed with fresh ECS. Fura-2-labeled TG neurons were immediately added to the cADDis-transfected primary cultured dental pulp slices. Thereafter, the co-culture was incubated in fresh ECS for 20 min before [Ca^2+^]_i_ and [cAMP]_i_ measurements.

### 2.7. Measurements of Intracellular cAMP- and/or Ca^2+^-Sensitive Dye Fluorescence

A dish containing cADDis-transfected dental pulp slices or a co-culture of fura-2-loaded TG neurons with cADDis-transfected dental pulp slices was mounted on the stage of a microscope (Olympus, Tokyo, Japan; IX73); this microscope had an HCImage system (Hamamatsu Photonic, Shizuoka, Japan), an excitation wavelength selector, and an intensified charge-coupled device camera system. The cADDis fluorescence emission was recorded at 560 nm in response to an excitation wavelength of 605 nm. Fura-2 fluorescence emission was recorded at 510 nm in response to alternating excitation wavelengths of 340 (F340) and 380 nm (F380). [Ca^2+^]_i_ was defined as the fluorescence ratio (R_F340/F380_) at the two excitation wavelengths, and the cADDis fluorescence and R_F340/F380_ of fura-2 were defined as the value of *F*, normalized to the resting value (*F*_0_), and as *F*/*F*_0_ units, respectively. All experiments were performed at 28 °C.

### 2.8. Mechanical Stimulation of the Single TG Neurons

Mechanical stimulation [12,13] was applied using a fire-polished glass micropipette with a tip diameter of 2–3 μm filled with ECS. The micropipette was pulled from the capillary tubes using a DMZ universal puller (Zeitz Instruments, Martinsried, Germany). The tip was positioned just above the target TG neuron, and the micropipette was moved vertically downward by 8.0 μm at a velocity of 2.0 μm/s [12,13] to generate a focal mechanical stimulation by the micromanipulator (μMp micromanipulator, Sensapex, Oulu, Finland) and software (Sensapex). The stimulation was applied for 22 s. Thereafter, the pipette was retracted at the same velocity. It is difficult to distinguish neurons from glial cells in TG cell cultures. To specifically reveal neuronal intracellular Ca^2+^ responses, responses in primary cultured TG cells were measured by the application of a solution containing a high concentration of K^+^ (50 mM K^+^), which induced membrane depolarization.

### 2.9. Measurement of the Intercellular Distance and Size of the Stimulated TG Neurons

Co-cultured odontoblasts in the dental pulp slice preparation and TG neurons were imaged using an intensified charge-coupled device camera (Hamamatsu Photonic) and microscope (Olympus). The distance from a mechanically stimulated TG neuron to each neighboring odontoblast and the size of the stimulated TG neuron were determined using the images (HCImage) by measuring the shortest distance of each pair of cells or the diameter of the stimulated TG neuron.

### 2.10. Mineralization Assay

Isolated odontoblasts were grown for 20–40 h in a basal medium and transferred to a mineralization medium (10 mM β-glycerophosphate and 50 μg/mL ascorbic acid in basal medium) for growth at 37 °C in 5% CO_2_. To determine the effects of CGRP activity on mineralization, odontoblasts were cultured in the mineralization medium without (as control) or with CGRP (rat) (50 nM) and with the CGRP inhibitor BIBN 4096 (0.1 nM) or the AC inhibitor SQ22536 (0.1 µM) for 7 days. During the 7-day culture period, the mineralization medium was changed twice per week. To detect the deposition of calcium and calcium phosphate, cells were subjected to alizarin red staining, and the mineralization efficiencies were measured using a microscope (Keyence, Osaka, Japan; X710). The regions of interest (ROIs) were determined for each odontoblast to measure the mean luminance intensity of the mineralized area in the total area (*I*) of the ROI. The mineralizing efficiencies were normalized and represented as *I*/*I*_0_ units, and the intensities (*I*) of alizarin red staining were normalized to the mean intensity area in the dental pulp (*I*_0_).

### 2.11. Statistical Analysis

Data are expressed as mean ± SE or SD of the mean of N observations, where N represents the number of experiments or cells tested. Non-parametric statistical significance was determined using the Friedman test and Mann–Whitney test with Dunn’s post-hoc test to assess the [cAMP]_i_ levels and mineralized areas in rat odontoblasts (Figures 2–6). Parametric statistical significance was determined using an unpaired t-test to analyze [Ca^2+^]_i_ in rat TG neurons (Figure 5). Statistical significance was set at *p* < 0.05. Statistical analyses were performed using GraphPad Prism 8.0 (GraphPad Software, La Jolla, CA, USA). The data were also analyzed using Origin 8.5 (OriginLab Corporation, Northampton, MA, USA).

## 3. Results

### 3.1. Expression of Heterotrimeric Gnas and CALCRL in Rat Odontoblasts

Primary cultured rat odontoblasts from the dental pulp slice preparations displayed immunoreactivity for DSPP (green in Figure 1A,C,D,F), heterotrimeric Gnas (red in Figure 1B,C), and CALCRL (as a component of CGRP receptor; red in Figure 1E,F).

### 3.2. Forskolin Dose-Dependently Increases the Intracellular cAMP Level

FSK was used as an AC activator to pharmacologically activate AC in rat odontoblasts. In the presence of external Ca^2+^ (2.5 mM), the application of six different concentrations of FSK (0.0001, 0.001, 0.01, 0.1, 2, and 13 μM) elicited rapid, transient, and concentration-dependent increases in [cAMP]_i_ in rat odontoblasts (Figure 2A–F). A semi-logarithmic plot (Figure 2G) shows the *F*/*F*_0_ values of cADDis fluorescence as a function of the applied FSK concentrations. The dependence of the changes in the *F*/*F*_0_ of [cAMP]_i_ induced by different concentrations of FSK was determined by fitting the data to the following function:*F*/*F*_0_ = [(*F*/*F*_0_min − *F*/*F*_0_max)/(1 + e ^(x−K)/dx)^)] + *F*/*F*_0_max,(1)
where K is the half-maximal concentration (50% effective concentration (EC_50_)) of FSK; x indicates the applied concentration of FSK; and *F*/*F*_0_max and *F*/*F*_0_min are the maximal and minimal *F*/*F*_0_ responses in [cAMP]_i_, respectively. The EC_50_ of FSK was 0.14 μM for the increases in [cAMP]_i_.

### 3.3. Repeated Application of Forskolin Has a Desensitizing Effect on the Increase in Intracellular cAMP Level

In the presence of extracellular Ca^2+^ (2.5 mM), the application of 0.1 μM FSK to odontoblasts led to a rapid and transient increase in [cAMP]_i_ (Figure 3A), with *F*/*F*_0_ units of [cAMP]_i_ reaching a peak value of 1.51 ± 0.04; this was followed by a rapid decay to near-baseline levels (*F*/*F*_0_ = 1). Repeated applications of 0.1 μM FSK decreased the amplitudes of FSK-evoked increases in [cAMP]_i_ level, and by the third application, a peak value of 1.22 ± 0.02 (N = 8; Figure 3B) was achieved for *F*/*F*_0_ units of [cAMP]_i_.

### 3.4. CGRP Increases the Intracellular cAMP Level

In the presence of extracellular Ca^2+^ (2.5 mM), the application of 50 nM of CGRP (rat) as a G_s_ protein-coupled CGRP receptor activator to odontoblasts induced a rapid and transient increase in [cAMP]_i_, with the *F*/*F*_0_ units reaching a peak value of 1.94 ± 0.1 (N = 9; Figure 4A) or 2.11 ± 0.11 (N = 11; Figure 4C); this was followed by a rapid decay to near-baseline levels (*F*/*F*_0_ = 1). CGRP-induced [cAMP]_i_ increases were significantly and reversibly inhibited by 0.1 nM BIBN 4096, a non-selective CGRP receptor inhibitor, to 1.21 ± 0.04 (N = 9; Figure 4B), and by 0.1 µM SQ22536, a pharmacological AC inhibitor, to 1.44 ± 0.04 (N = 11; Figure 4D) *F*/*F*_0_ units.

### 3.5. Direct Mechanical Stimulation of the TG Neurons Simultaneously Increases the [Ca^2+^]_i_ in the Neurons and the [cAMP]_i_ in the Odontoblasts Approximating the Stimulated Neurons

Based on simultaneous measurements of [cAMP]_i_ in the odontoblasts and [Ca^2+^]_i_ in the neurons, in the co-culture system with external 2.5 mM Ca^2+^, the application of focal and direct mechanical stimulation to single TG neurons (8.0 μm in depth) induced transient increases in [Ca^2+^]_i_ (green lines, Figure 5A,B) to a peak *F*/*F*_0_ value of 3.23 ± 2.45 (N = 3; Figure 5A) in the control condition. Transient increases in [cAMP]_i_ (red lines, Figure 5A) were also observed in neighboring odontoblasts during the direct mechanical stimulation of TG neurons (N = 27; Figure 5A). The amplitude of the [cAMP]_i_ increase in neighboring odontoblasts reduced with increase in their distance from the mechanically stimulated TG neurons (N = 27; Figure 5A and open columns of Figure 5C). Furthermore, a time delay in the increase in [cAMP]_i_ was observed in the nearby odontoblasts during the mechanical stimulation of single TG neurons (time when the mechanical stimulation applied to the stimulated TG neuron was set to 0 s; vertical dotted line, Figure 5A,B). In the presence of 0.1 nM BIBN 4096, a non-selective CGRP receptor inhibitor, the mechanical stimulation of TG neurons induced [Ca^2+^]_i_ increases, reaching a peak *F*/*F*_0_ value of 2.40 ± 0.87 (N = 3; Figure 5B); however, the [cAMP]_i_ increases in the neighboring odontoblasts were significantly inhibited (N = 42; Figure 5B) compared to the increases in those without BIBN 4096. There were no significant differences in the *F*/*F*_0_ values of the mechanically stimulated [Ca^2+^]_i_ increases in the presence and absence of external BIBN 4096. After measuring [Ca^2+^]_i_ and [cAMP]_i_ in the co-cultured cells, immunofluorescence analyses were performed to identify the nature of the stimulated TG neurons. TG cells were identified by the application of a solution containing a high concentration of K^+^ (50 mM), which induced membrane depolarization to specifically reveal neuronal intracellular Ca^2+^ responses in primary culture.

The mechanically stimulated TG neurons (arrowheads) displayed immunopositivity to CGRP (red) but not to NF-H (blue) (Figure 5D–F). The size of the stimulated TG neurons, which was measured before mechanical stimulation, was 15.34 ± 2.59 µm (N = 6).

### 3.6. CGRP–CGRP Receptor Signaling Regulates Defensive Dentin Demineralization

We investigated the effects of CGRP activity on mineralization in isolated odontoblasts. Alizarin red staining (Figure 6A–D) indicates the mineralization levels based on the staining intensity (see Materials and Methods), represented as *I*/*I*_0_ units; the intensities (*I*) of the stains were normalized to the mean intensity area in the dental pulp (*I*_0_; Figure 6E). The application of 50 nM CGRP (rat) significantly reduced the mineralization levels compared to the control level and that with the CGRP receptor antagonist (0.1 nM), BIBN 4096, or the AC inhibitor (0.1 μM) SQ22536.

## 4. Discussion

According to the findings of the present study, DSPP-positive rat odontoblasts express Gα_s_-protein-coupled CGRP receptors. CGRP receptor activation was found to regulate the AC signal transduction pathway to produce [cAMP]_i_. The increases in [cAMP]_i_ levels were decreased by repeated stimulation of AC. AC produces cAMP from adenosine triphosphate (ATP), and the depletion of ATP may cause a decrease in [cAMP]_i_ levels through repeated AC activation. We identified odontoblasts based on their expression of DSPP, a mature odontoblast marker [27,28]. To record cAMP appropriately at the single-cell level, gap junctions comprising members of the connexin family between odontoblasts should be disconnected, because gap junctional communication allows intercellular communication via intracellular cAMP/Ca^2+^. Therefore, we prepared a single odontoblast level. We confirmed that cells retained the high protein expression level of DSPP and preserved their function even when the cells were growing out. The mechanical stimulation of TG neurons induced not only [Ca^2+^]_i_ increases in the stimulated neurons but also [cAMP]_i_ increases in the odontoblasts near the stimulated TG neurons. The increase in [cAMP]_i_ in the neighboring odontoblasts was inversely proportional to their distance from the mechanically stimulated TG neurons, suggesting that diffusible substances are released from mechanically stimulated TG neurons. Moreover, the increase in [cAMP]_i_ in nearby odontoblasts during neuronal mechanical stimulation was significantly suppressed by the application of BIBN 4096, a CGRP receptor antagonist. These results functionally align with previous results that revealed the expression of CGRP receptors by odontoblasts [29] and the function of released CGRP from mechanically stimulated TG neurons as a mediator that contributes to the development of the axon reflex in dental pulp tissue [11].

After [Ca^2+^]_i_ and [cAMP]_i_ measurements in TG neurons and odontoblasts in the co-culture, mechanically stimulated small-sized TG neurons (15.34 ± 2.59 µm in diameter) were observed via immunofluorescence and identified to exhibit CGRP positivity and NF-H negativity. TG neurons are subclassified into nine cell types, and these cells are divided into two groups: small-sized (15–24 µm) and medium-sized (25–38 µm) [30]. NF-H is a marker for medium- to large-sized A-fiber neurons, whereas peptidergic C-fiber neurons express CGRP [31,32]. Therefore, the cells mechanically stimulated in the present study were peptidergic C-fiber neurons. CGRP is released from the terminals of pulpal nociceptors, which comprise unmyelinated C fibers [7]. Furthermore, CGRP localizes in TG neurons and nerve terminals in the dental pulp [31] and contributes to vasodilation via the axon reflex in pulpitis [7]. Nerve fibers in dental pulp express CGRP [33] and dental pulp cells express CGRP receptors increasingly during inflammatory phenomena such as acute irreversible pulpitis [34]. In this study, endogenous CGRP expression could not be observed in physiological conditions of odontoblasts using immunostaining (personal communication with TO); therefore, further study will be needed to reveal whether the inflammatory response induces endogenous CGRP expression in odontoblasts. Thus, CGRP released from peptidergic C neurons is an intercellular mediator among the cells in dental pulp, including odontoblasts, and this may imply that CGRP released from C neurons mediates neurogenic inflammation in dental pulp.

The mechanical stimulation of nerve fibers by pulsating vessels activates mechanosensitive Piezo channels and stimulates neuronal CGRP release, which contributes to neurogenic inflammation. The activation of Piezo2 channels, which are mechanosensitive ion channels, promotes CGRP release from TG neurons during migraine attacks [35]. Moreover, medium-sized primary afferent neurons innervating the dental pulp express Piezo2 channels, as they are low-threshold mechanoreceptors [36]. Although a further study is required, previous results suggest that the increases in [Ca^2+^]_i_ induced by mechanical stimulation are mediated by mechanosensitive Piezo channel activation in TG neurons. However, the functional roles of CGRP in the cellular functions of odontoblasts, the essential player in dentinogenesis, are unclear. A previous study demonstrated that CGRP produced by stimulated nociceptive neurons plays a role in murine molar dental pulp stem cell proliferation, differentiation, and inflammatory gene expression [37]. By contrast, it has been reported to inhibit bone mineralization in osteoblasts [38,39]. Our findings revealed that CGRP significantly decreased mineralization levels. Mineralization in odontoblasts which indicated positivity for DSPP (specific late odontoblast differentiation marker) [27] cultured in a mineralization medium without CGRP (as a control experiment) mimicked the physiological or developmental conditions of dentinogenesis. Furthermore, the application of CGRP with a CGRP receptor antagonist or an AC inhibitor led to the recovery of mineralization levels compared to single CGRP application. Our data suggest that the neuropeptide CGRP, which is released from TG neurons, might function to prevent excessive mineralization and avoid increasing pressure. Based on these results, CGRP released from peptidergic C neurons in the axon reflex suppresses dentin regeneration to prevent internal tissue pressure increases. Of note, a reduction in the volume of the dental pulp chamber by excessive pathological dentin formation results in an increase in tissue pressure caused by pulpal inflammatory responses, which may accelerate the responses to dental pulp inflammation. Thus, a CGRP-induced reduction in mineralization efficiency by odontoblasts may be essential for defensive reactions during dental pulp inflammation.

In conclusion, we revealed the functional expression of Gα_s_ protein-coupled CGRP receptors in DSPP-immunopositive odontoblasts. CGRP receptor activation increases [cAMP]_i_ by activating AC. The mechanical stimulation of peptidergic C neurons of the TG, which mimics mechanical stimulation due to dental pulp inflammation, induces CGRP release following mechanosensitive ion channel activation. CGRP released from peptidergic C neurons increases [cAMP]_i_ in odontoblasts via CGRP receptor activation and reduces dentin mineralization. Thus, the CGRP–CGRP receptor axis plays a critical role in the regulation of dentinogenesis via intercellular communication. These results may provide functional evidence regarding the axon reflex mediated via the CGRP–CGRP receptor axis in dental pulp.

## Figures and Tables

**Figure 1 biomolecules-12-01747-f001:**
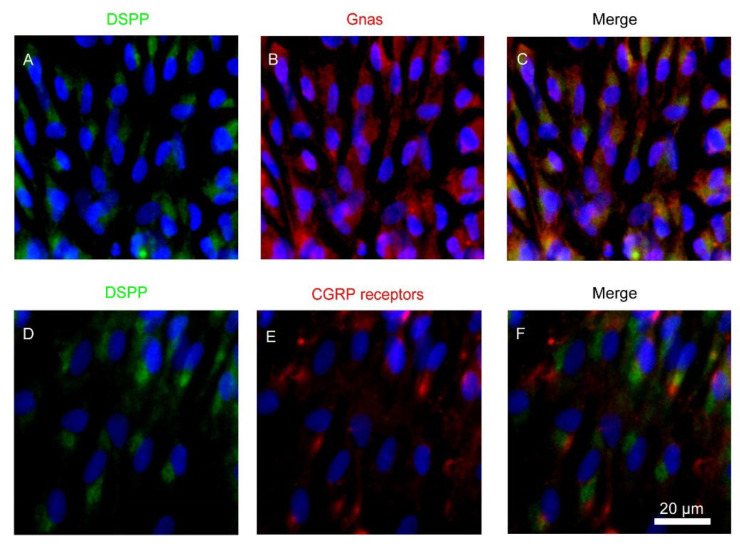
Expression of the heterotrimeric G-protein α-subunit Gα_s_ (Gnas) and calcitonin receptor-like receptor (CALCRL) in rat odontoblasts. (**A**–**F**) Odontoblasts in dental pulp slice preparations showed positive immunoreactivity to DSPP (green in **A**,**C**,**D**,**F**), Gnas (red in **B**,**C**), and calcitonin receptor-like receptor (CALCRL) (red in **E**,**F**). Nuclei are colored blue. Scale bar: 20 μm. No fluorescence was detected in the negative controls (not shown). Data were obtained using a 40× magnification objective lens under a microscope. Enlarged images are shown as representative areas.

**Figure 2 biomolecules-12-01747-f002:**
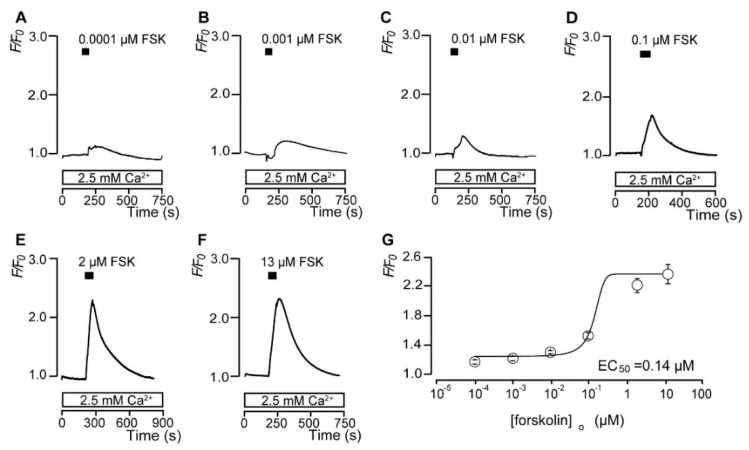
Forskolin dose−dependently increases the intracellular cAMP level ([cAMP]_i_). (**A**–**F**) Representative traces of the transient increases in [cAMP]_i_ during the administration of different concentrations of forskolin (FSK) (**A**: 0.0001 μM; **B**: 0.001 μM; **C**: 0.01 μM; **D**: 0.1 μM; **E**: 2 μM; **F**: 13 μM) in the presence of extracellular Ca^2+^ (2.5 mM) (white boxes at the bottom). Black boxes indicate the time periods of FSK addition to the standard extracellular solution. (**G**) The data points illustrate the *F*/*F*_0_ values in [cAMP]_i_ as a function of the applied FSK concentration. Each point represents the mean ± SE. The curve on the semi-logarithmic scale was fitted according to Equation (1), which is described in the text.

**Figure 3 biomolecules-12-01747-f003:**
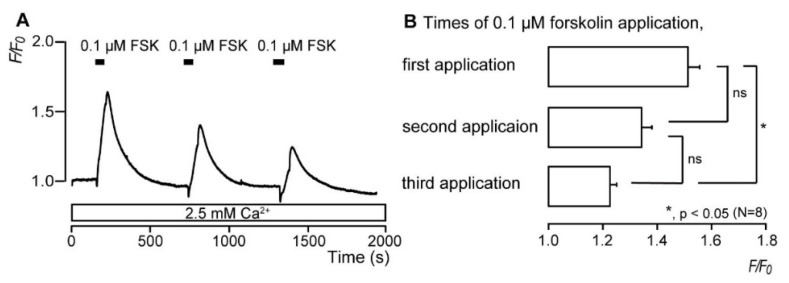
Repeated application of forskolin elicits a desensitizing effect on the intracellular cAMP level ([cAMP]_i_). (**A**) Representative trace of the transient increases in [cAMP]_i_ during the administration of 0.1 μM forskolin (FSK) in standard extracellular solution with 2.5 mM extracellular Ca^2+^ (white box at the bottom). Black boxes indicate the time periods of FSK addition to the standard extracellular solution. (**B**) Summary bar graph represents values of *F*/*F*_0_ for the increases in [cAMP]_i_ after first (upper), second (middle), and third (lower) applications of 0.1 μM FSK. Each bar denotes the mean ± SE of 8 experiments. Asterisks denote statistically significant differences between columns (shown by solid lines): * *p* < 0.05.

**Figure 4 biomolecules-12-01747-f004:**
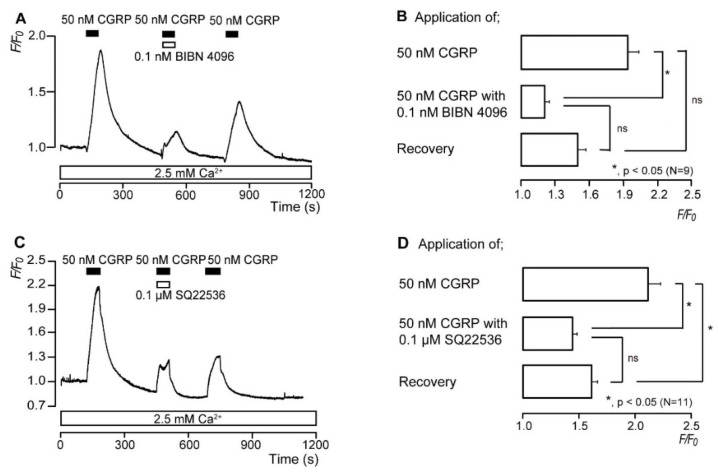
CGRP increases the intracellular cAMP level ([cAMP]_i_). (**A**,**C**) Representative traces of the transient [cAMP]_I_ level increases in response to 50 nM CGRP (rat) with or without 0.1 nM BIBN 4096 (**A**) or 0.1 μM SQ22536 (**C**) in the presence of extracellular Ca^2+^ (2.5 mM) (white boxes at the bottom). Black boxes indicate the time periods of CGRP (rat) application to the extracellular solution. White boxes at the top indicate the time of addition of BIBN 4096 (**A**) or SQ22536 (**C**) to the extracellular solution. (**B**,**D**) Summary bar graphs show CGRP (rat)-induced [cAMP]_i_ level increases in the control (upper column) or with (middle column) 0.1 nM BIBN 4096 (**B**) or 0.1 μM SQ22536 (**D**) in the presence of extracellular Ca^2+^ (2.5 mM). Each recovery effect (lower column in **B**,**D**) shows the reversible effect of BIBN 4096 (**B**) and SQ22536 (**D**). Each bar denotes the mean ± SE. Numbers in parentheses show the number of experiments. Asterisks denote statistically significant differences between columns (shown by solid lines): * *p* < 0.05.

**Figure 5 biomolecules-12-01747-f005:**
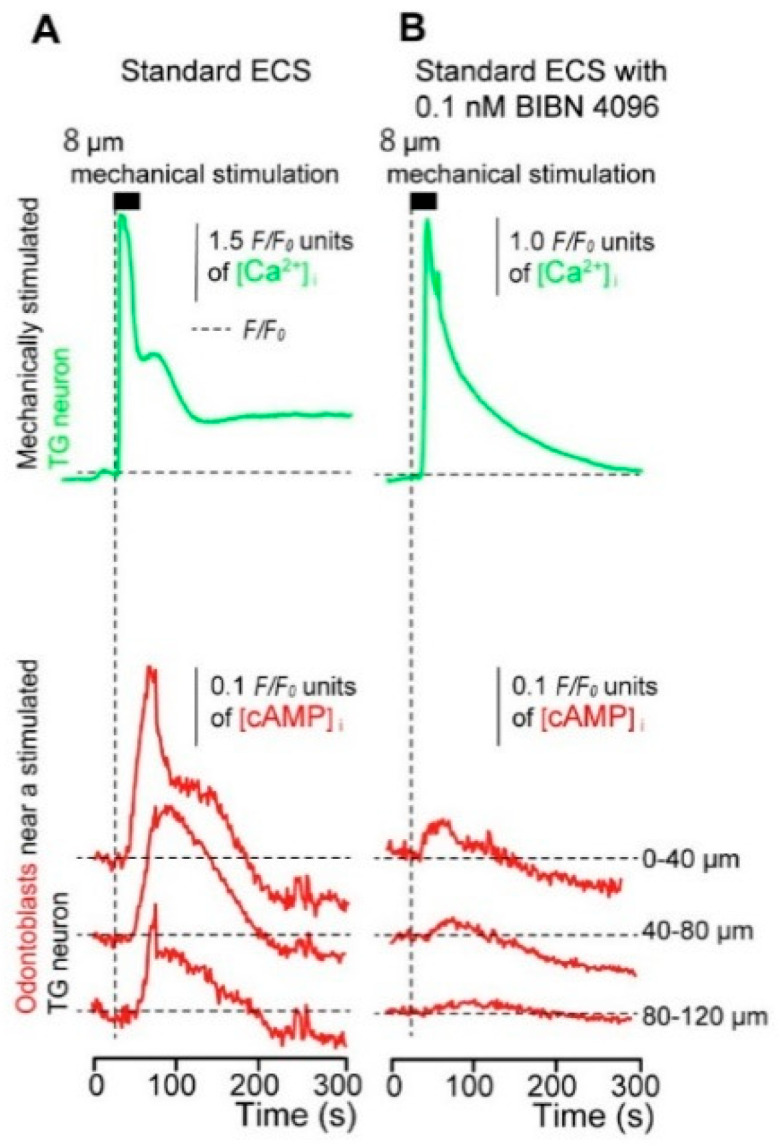
Direct mechanical stimulation of TG neurons simultaneously increases [Ca^2+^]_i_ in neurons and [cAMP]_i_ in the neighboring odontoblasts. (**A**,**B**) Representative traces showing transiently increased [Ca^2+^]_i_ in a mechanically stimulated TG neuron (green lines) and [cAMP]_i_ level in neighboring odontoblasts (red lines) following focal and direct mechanical stimulation by a micropipette under standard extracellular solution. The traces shown are without (**A**) or with 0.1 nM BIBN 4096 (**B**). Horizontal dotted lines indicate the baseline (*F*/*F*_0_ = 1.0) for each response, while vertical dotted lines represent the application time of the mechanical stimulation. The black boxes at the top show the timing of the mechanical stimulation based on the displacement of a micropipette to a depth of 8 μm. The responses from the nearby odontoblasts were recorded from cells located 0–120 μm away from the stimulated TG neuron. The distance of each cell from the mechanically stimulated TG neuron is indicated on the right side of each trace (**C**). The *F*/*F*_0_ values of neighboring odontoblasts located within 0–40 μm, 41–80 μm, and 81–120 μm from the stimulated TG neuron in standard extracellular solution (open columns) or standard extracellular solution with BIBN 4096 (0.1 nM) (red columns) are given. The [cAMP]_i_ increases in the neighboring odontoblasts reduced with the increase in their distance from the mechanically stimulated TG neuron with and without BIBN 4096. Each bar denotes the mean ± SE. Asterisks denote statistically significant differences between columns or values (shown by solid lines): * *p* < 0.05. (**D**–**F**) Mechanically stimulated TG neuron showing positive immunoreactivity for CGRP (**D**,**F**) and negative immunoreactivity for NF-H (**E**,**F**). Nuclei are indicated by a gray color. Scale bar: 20 μm. Arrowheads indicate a mechanically stimulated cell. No fluorescence was detected in the negative controls (not shown). Data were obtained using a 120× magnification objective lens under a microscope. Enlarged images are shown as representative areas.

**Figure 6 biomolecules-12-01747-f006:**
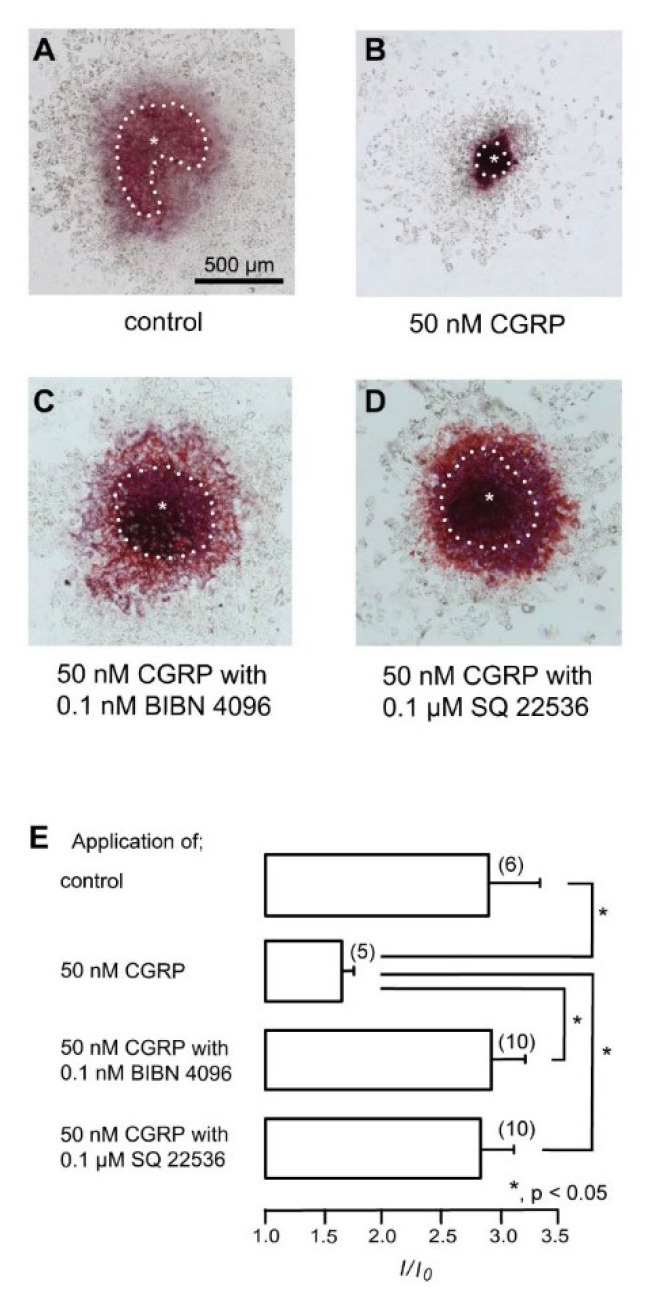
CGRP receptor signaling regulates the defensive reaction of odontoblasts. (**A**–**D**) Isolated rat dental pulp slices were cultured for 7 days in a mineralization medium without pharmacological intervention (**A**) or with 50 nM CGRP (**B**), 50 nM CGRP with 0.1 nM BIBN 4096 (**C**), or 50 nM CGRP with 0.1 μM SQ22536 (**D**) at pH 7.4 and stained using Alizarin red (red, calcium deposition). White dotted lines indicate the borderline between mineralized odontoblasts and dental pulp. Asterisks indicate the dental pulp area. Data were obtained using a 20× magnification objective lens under a microscope. Each dataset was tiled with 63 photos to acquire the constructed data. To obtain data on the mineralized area constituted by odontoblasts, the mineralized area in the total area (*I*) was divided by the mineralized area of the dental pulp (*I*_0_). (**E**) The estimated mineralization levels were 2.89 ± 0.43 *I*/*I*_0_ in the absence of CGRP receptor modifiers (as controls), 1.64 ± 0.1 *I*/*I*_0_ with 50 nM CGRP, 2.91 ± 0.29 *I*/*I*_0_ with 50 nM CGRP and 0.1 nM BIBN 4096, and 2.82 ± 0.29 *I*/*I*_0_ with 50 nM CGRP and 0.1 μM SQ22536. Each column denotes the mean ± SE of each experiment. Statistically significant differences between columns (shown by solid lines) are indicated by asterisks. * *p* < 0.05.

## Data Availability

All data are presented in the article.

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
