# Peer review of "Gαs-Coupled CGRP Receptor Signaling Axis from the Trigeminal Ganglion Neuron to Odontoblast Negatively Regulates Dentin Mineralization"

_biomolecules, 2022, doi:10.3390/biom12121747_

Round 1
Reviewer 1 Report
The manuscript entitled “Gαs-coupled CGRP receptor signaling axis from the trigeminal ganglion neuron to odontoblast negatively regulates dentin mineralization (manuscript ID: biomolecules-1978771)” was described by Drs. Natsuki Saito et al. The authors found that intercellular communications between odontoblasts/dental pulp cells and trigeminal ganglion (TG) neurons regulate dental pulp cells/odontoblast mineralization via the intracellular calcitonin gene-related peptide (CGRP) receptor and cyclic adenosine monophosphate (cAMP) signaling pathway. In cocultured dental pulp slice and TG neurons, the neurons influence cAMP release from odontoblasts. Based on these studies, the authors showed that CGRP inhibits dental pulp cell/odontoblast cell mineralization. However, these are several comments as follows:
1. It is known that CGRP is broadly expressed in a lot of cells and tissues such as dental pulp cells, odontoblasts, and neurons. The CGRP peptide binds to CGRP receptors, forming the complex. The complex activates adenylate cyclase (AC), which stimulates ATP to cAMP. cAMP signal pathway regulates downstream gene expression and ion activity such as calcium (Ca++). In a lot of reports, Increased Ca++ induces dental pulp cell/odontoblast differentiation in vitro and in vivo. However, in this study, it showed that CGRP inhibits odontoblast/dental pulp cell mineralization. In the discussion, the authors need to discuss the issue.
2. Although TN neurons express and secrete CGRP in normal conditions and inflammatory responses, CGRP can influence neighboring dental pulp cell and odontoblast physiological and pathological activities. However, during injury and inflammation such as pulpitis, dental pulp cells/odontoblasts increasingly express endogenous CGRP (References 1 and 2). Besides the TN neuron system, the dental pulp cells/odontoblasts have capable of protective systems responsible for injury and inflammation. Did the authors measure cAMP or CGRP activity in the dental pulp slice during machinal stimulation in this study?
3. In dental pulp slices, besides cell morphology and positions, how do the authors identify which are odontoblasts and dental pulp cells? Does CGRP influence odontoblast or dental pulp cell differentiation? Does CGRP only inhibit dental pulp cells or odontoblasts or both?
4. Few papers reported that CGRP induces dental pulp cell/odontoblast differentiation and dentin regeneration (References 3 and 4). The authors need to discuss this in the discussion section.
5. What mechanisms does CGRP inhibit dental pulp cell/odontoblast mineralization? Does CGRP downregulate the expression of dentin sialophosphoprotein (DSPP) or dentin matrix protein (DMP1), which are important extracellular matrix proteins during dentin development and regeneration?
References
1. Fehrenbacher JC, Sun XX, Locke EE, Henry MA, Hargreaves KM. Capsaicin-evoked iCGRP release from human dental pulp: a model system for the study of peripheral neuropeptide secretion in normal healthy tissue. Pain. 2009 Aug;144(3):253-261.
2. Caviedes-Bucheli J, Arenas N, Guiza O, Moncada NA, Moreno GC, Diaz E, Munoz HR. Calcitonin gene-related peptide receptor expression in healthy and inflamed human pulp tissue. Int Endod J. 2005 Oct;38(10):712-7.
3. Kline LW, Yu DC. Effects of calcitonin, calcitonin gene-related peptide, human recombinant bone morphogenetic protein-2, and parathyroid hormone-related protein on endodontically treated ferret canines. J Endod. 2009 Jun;35(6):866-9.
4. Moore ER, Michot B, Erdogan O, Ba A, Gibbs JL, Yang Y. CGRP and Shh Mediate the Dental Pulp Cell Response to Neuron Stimulation. J Dent Res. 2022 Aug;101(9):1119-1126. doi: 10.1177/00220345221086858.
Author Response
Dear Editor and Reviewer 1,
We deeply appreciate the constructive and productive comments and suggestions of the editor and reviewers. Thank you for reviewing our manuscript and offering valuable advice. We have addressed your comments in a point-by-point format and have revised the manuscript throughout.
Reviewer 1:
Comment 1
The manuscript entitled “Gαs-coupled CGRP receptor signaling axis from the trigeminal ganglion neuron to odontoblast negatively regulates dentin mineralization (manuscript ID: biomolecules-1978771)” was described by Drs. Natsuki Saito et al. The authors found that intercellular communications between odontoblasts/dental pulp cells and trigeminal ganglion (TG) neurons regulate dental pulp cells/odontoblast mineralization via the intracellular calcitonin gene-related peptide (CGRP) receptor and cyclic adenosine monophosphate (cAMP) signaling pathway. In cocultured dental pulp slice and TG neurons, the neurons influence cAMP release from odontoblasts. Based on these studies, the authors showed that CGRP inhibits dental pulp cell/odontoblast cell mineralization. However, these are several comments as follows:
It is known that CGRP is broadly expressed in a lot of cells and tissues such as dental pulp cells, odontoblasts, and neurons. The CGRP peptide binds to CGRP receptors, forming the complex. The complex activates adenylate cyclase (AC), which stimulates ATP to cAMP. cAMP signal pathway regulates downstream gene expression and ion activity such as calcium (Ca++). In a lot of reports,Increased Ca++ induces dental pulp cell/odontoblast differentiation in vitro and in vivo. However, in this study, it showed that CGRP inhibits odontoblast/dental pulp cell mineralization. In the discussion, the authors need to discuss the issue.
Answer
Thank you for pointing this out to us. Dental pulp has low compliance due to the surrounding hard tissues, such as dentin and enamel. Our data revealed that CGRP induced the downregulation of mineralization, as shown in Fig. 6. CGRP is a well-known neuropeptide involved in inflammation and the axon reflex. Our data suggest that CGRP may function to prevent excessive mineralization and thereby avoid increasing pressure. We have included this explanation in the discussion section. Please find the yellow highlights on page 15, line 465 to page 15, line 467 in the revised manuscript.
From page 15, line 465 to page 15, line 467
Our data suggest that the neuropeptide CGRP, which is released from TG neurons, might function to prevent excessive mineralization and avoid increasing pressure.
Reviewer 1’s explanation has been corrected. CGRP is expressed in several tissues in other organs, and CGRP regulates physiological and pathological functions. We have learned that Villa et al., reported that CGRP inhibits osteoprotegerin production in human osteoblast-like cells via the cAMP/PKA-dependent pathway. CGRP can modulate the balance between osteoblast and osteoclast activities (Villa et al., Am J Physiol Cell Physiol. 2006). Another group has reported that CGRP and brain-derived serotonin are related to bone loss in ovariectomized rats (Zhang et al., Brain Res Bull, 2021). Moore et al. recently reported that dental pulp cells activate their mineralization ability via CGRP and Shh (Moore et al., J Dent Res, 2022). Thus, CGRP, its responder, and its regulatory signaling pathways are still ambiguous in their mineralization ability in different tissue cells.
At this point, we have added two articles to the references and inserted additional explanations from page 15, line 455 to page 15, line 459 with yellow highlights in the revised manuscript.
From page 15, line 455 to page 15, line 459
A previous study demonstrated that CGRP produced by stimulated nociceptive neurons plays a role in murine molar dental pulp stem cell proliferation, differentiation, and inflammatory gene expression [35]. By contrast, it has been reported to inhibit bone mineralization in osteoblasts [36, 37].
From page 17, line 600 to page 17, line 605
36.Villa, I.; Mrak, E.; Rubinacci, A.; Ravasi, F.; Guidobono, F. CGRP Inhibits Osteoprotegerin Production in Human Osteoblast-like Cells via CAMP/PKA-Dependent Pathway. Am J Physiol Cell Physiol 2006, 291, C529-537, DOI:10.1152/ajpcell.00354.2005.
37.Zhang, R.-H.; Zhang, X.-B.; Lu, Y.-B.; Hu, Y.-C.; Chen, X.-Y.; Yu, D.-C.; Shi, J.-T.; Yuan, W.-H.; Wang, J.; Zhou, H.-Y. Calcitonin Gene-Related Peptide and Brain-Derived Serotonin Are Related to Bone Loss in Ovariectomized Rats. Brain Res Bull 2021, 176, 85–92, DOI:10.1016/j.brainresbull.2021.08.007
Comment 2
Although TN neurons express and secrete CGRP in normal conditions and inflammatory responses, CGRP can influence neighboring dental pulp cell and odontoblast physiological and pathological activities. However, during injury and inflammation such as pulpitis, dental pulp cells/odontoblasts increasingly express endogenous CGRP (References 1 and 2). Besides the TN neuron system, the dental pulp cells/odontoblasts have capable of protective systems responsible for injury and inflammation. Did the authors measure cAMP or CGRP activity in the dental pulp slice during machinal stimulation in this study?
- Fehrenbacher JC, Sun XX, Locke EE, Henry MA, Hargreaves KM. Capsaicin-evoked iCGRP release from human dental pulp: a model system for the study of peripheral neuropeptide secretion in normal healthy tissue. Pain. 2009 Aug;144(3):253-261.
- Caviedes-Bucheli J, Arenas N, Guiza O, Moncada NA, Moreno GC, Diaz E, Munoz HR. Calcitonin gene-related peptide receptor expression in healthy and inflamed human pulp tissue. Int Endod J. 2005 Oct;38(10):712-7.
Answer
Thank you for your valuable comments. No, we did not measure CGRP activity in odontoblasts by mechanical stimulation. In our present study, CGRP was negative in odontoblasts as shown in this letter. Immunostaining of co-cultured cells composed of TG cells and odontoblasts revealed that endogenous CGRP expression was not activated in odontoblasts by TG co-culture itself (personal communication by TO). We inserted this discussion in the revised manuscript (page 14, line 438 to page 14, line 441).
From page 14, line 438 to page 14, line 441
In this study, immunostaining of co-cultured cells composed of TG cells and odontoblasts revealed that endogenous CGRP expression was not activated in odontoblasts by TG co-culture (personal communication with TO).
As reviewer 1 suggested, we have also considered that we need to evaluate the functional differences in mechanosensitivity and mechanosensitive cAMP signaling between dental pulp (stem/stromal) cells and odontoblasts. Indeed, our group is currently addressing these valuable comments and is preparing the next paper.
Immunofluorescence staining of co-cultured cells revealed that CGRP was not expressed in odontoblasts.
The left photograph shows co-cultured cells composed of both odontoblasts and TG neurons. The image on the right is the magnified area outlined by the rectangle. CGRP-positive cells were negative for IB4, which is a non-peptidergic neuronal biomarker (Cheryl et al., J neurosci, 1999). White arrowheads indicate odontoblasts and yellow arrowheads indicate TG neurons.
Comment 3
In dental pulp slices, besides cell morphology and positions, 1) how do the authors identify which are odontoblasts and dental pulp cells? 2) Does CGRP influence odontoblast or dental pulp cell differentiation? 3) Does CGRP only inhibit dental pulp cells or odontoblasts or both?
Answer
Thank you for pointing this out to us. We have addressed this point by point. To answer the first question, we identified the odontoblast layer in slice samples using the expression of DSPP (Krivanek et al., Nat Commun, 2020). As shown in this letter, the odontoblastic layer strongly expresses DSPP in most outer layers compared with the central area, which is composed of dental pulp stem/stromal cells. Krivanek et al. reported that DSPP was enriched in mature odontoblasts at single-cell sequence resolution. In our study, after peeling out the dentin during the preparation of odontoblast slices, odontoblasts were released from internal pressure. Therefore, some odontoblasts grew out of the central dental pulp area. This step is quite important to make us record cAMP appropriately under the condition that gap junctions comprising members of the connexin family between odontoblasts are disconnected. Therefore, we prepared a single odontoblast level. We confirmed that cells retained the high protein expression level of DSPP and also conserved their function even when cells were growing out (see below, photo in this response). We inserted an explanation from page 14, line 410 to page 14, line 416 in the revised manuscript.
From page 14, line 410 to page 14, line 416
We identified odontoblasts based on their expression of DSPP, a mature odontoblast marker [27,28]. To record cAMP appropriately at the single-cell level, gap junctions comprising members of the connexin family between odontoblasts should be disconnected, because gap junctional communication allows intercellular communication via intracellular cAMP/Ca2+. Therefore, we prepared a single odontoblast level. We confirmed that cells retained the high protein expression level of DSPP and preserved their function even when the cells were growing out.
Immunofluorescence staining of dental pulp slices revealed that the mature odontoblast marker, DSPP, was strongly expressed in most outer layers compared with the central area, which was composed of dental pulp stem/stromal cells in slice samples. The asterisk indicates the area composed of dental pulp stem/stromal cells.
To questions 2 and 3
The paper focused on odontoblast cAMP signaling, but not dental pulp cells. We did not evaluate the difference between dental pulp cells and odontoblasts. Our group is currently addressing these valuable comments and is preparing the next paper.
Comment 4
Few papers reported that CGRP induces dental pulp cell/odontoblast differentiation and dentin regeneration (References 3 and 4). The authors need to discuss this in the discussion section.
- Kline LW, Yu DC. Effects of calcitonin, calcitonin gene-related peptide, human recombinant bone morphogenetic protein-2, and parathyroid hormone-related protein on endodontically treated ferret canines. J Endod. 2009 Jun;35(6):866-9.
- Moore ER, Michot B, Erdogan O, Ba A, Gibbs JL, Yang Y. CGRP and Shh Mediate the Dental Pulp Cell Response to Neuron Stimulation. J Dent Res. 2022 Aug;101(9):1119-1126. doi: 10.1177/00220345221086858.
Answer
Thank you for this suggestion. As described above, we feel that this is a very interesting result of our study. Once again, we think that the neuropeptide CGRP might function to prevent excessive mineralization thereby an avoiding undesirable increase in pressure. We believe that an in vivo assay is essential to clarify this. In this regard, we have added additional explanations from page 15, line 455 to page 15, line 459 with yellow highlights in the revised manuscript.
From page 15, line 455 to page 15, line 459
A previous study demonstrated that CGRP produced by stimulated nociceptive neurons plays a role in murine molar dental pulp stem cell proliferation, differentiation, and inflammatory gene expression [35]. By contrast, it has been reported to inhibit bone mineralization in osteoblasts [36, 37].
Comment 5
What mechanisms does CGRP inhibit dental pulp cell/odontoblast mineralization? Does CGRP downregulate the expression of dentin sialophosphoprotein (DSPP) or dentin matrix protein (DMP1), which are important extracellular matrix proteins during dentin development and regeneration?
Answer
Thank you for pointing this out to us. Indeed, this is an interesting question. We have not focused on the reviewer’s details. 1 suggested. We previously reported that the mechanosensitive ion channel, Piezo1, suppresses the mineralization of odontoblasts (Matsunaga et al., Front Physiol, 2021). Mineralization caused by external factors (including physicochemical and ligands) requires attention not only to understand protein and mRNA expression changes or protein levels but also to intracellular signaling pathways, which is an intriguing topic for future research.
Reviewer 2 Report
This manuscript presents a very interesting study showing that sensory neurons trough the release of CGRP regulate odontoblast activity and decrease dentin formation.
The manuscript is well written and provides new data on the role of sensory neuron in dental pulp physiology.
Comments:
The Authors investigated the interaction between sensory neurons and odontoblasts in vitro and used a model of dental pulp explant containing odontoblast. However, it seems that in some experiments isolated odontoblasts where used.
Did the Authors used primary culture of odontoblasts? if so could they provide the detailed methods.
Could the authors clarify what model they used for each experiment?
When using pulp slice model for cAMP assay how did the authors confirmed the recorded cells were actually odontoblast and not another cell type present in the pulp tissue?
The figure 1 shows high magnification images of co-staining of DSPP with CALCRL/Gnas in dental pulp slices. It looks that odontoblasts are spread over a large area and do not form a single layer of cells as expected. And illustration at low magnification, showing all the pulp tissue, to confirm the location of the DSPP staining could help.
The discussion and conclusion put this study in the context of pulp inflammation. Pulp inflammation generally happens when the tooth is developed, but the present study used pups rats tissue in which the tooth is likely not fully developed. As biological mechanisms can be different during the development and in adults, do the authors assume the mechanism they describe applies to adults teeth? Some references supporting that pups rat teeth are physiologically similar to adult rat teeth could be helpful to support the conclusion.
Author Response
Dear Editor and Reviewer 2,
We deeply appreciate the constructive comments and suggestions of the editor and Reviewer 2. Thank you for reviewing our manuscript and offering your valuable advice. We have substantially addressed your comments with point-by-point responses and have revised the manuscript accordingly.
Reviewer 2:
Comment 1
The Authors investigated the interaction between sensory neurons and odontoblasts in vitro and used a model of dental pulp explant containing odontoblast. However, it seems that in some experiments isolated odontoblasts where used. Did the Authors used primary culture of odontoblasts? if so could they provide the detailed methods. Could the authors clarify what model they used for each experiment?
Answer
Thank you for pointing this out to us. In this study, we used primary cultured odontoblasts. This method has been well discussed in cited papers, which were reported by our group and others. After peeling out the dentin during the preparation of odontoblast slices, odontoblasts are released from internal pressure. Therefore, some odontoblasts grew out of the central dental pulp area. This step is important to record cAMP appropriately under the condition that gap junctions comprising members of the connexin family between odontoblasts are disconnected. Therefore, we prepared a single odontoblast level. We confirmed that cells conserved the high protein expression level of DSPP (see the below photo in this letter), which is known as a mature odontoblast marker and also conserved their function even when cells were growing out. Our group has reported this method in articles published over the past 25 years.
Shibukawa, Suzuki, Bull Tokyo Dent Coll, 1997
Shibukawa, Suzuki, J Bone Miner Res, 2003
Okumura et al., Arch Histol Cytol, 2005
Son et al., J Dent Res, 2009
Tsumura et al., J Endod, 2010
Magloire et al., J Orofac Pain, 2010
Tsumura et al., Cell Calcium, 2012
Sato et al., J Endod, 2013
Tsumura et al., PLoS One, 2013
Shibukawa et al., Pflügers Archiv, 2015
Tokuda et al., Med. Hypotheses, 2015
Sato et al., J Endod, 2018
Kimura et al., Front Physiol, 2018
Immunofluorescence staining of dental pulp slices revealed that the mature odontoblast marker, DSPP, was strongly expressed in most outer layers compared with the central area, which was composed of dental pulp stem/stromal cells in slice samples. The asterisk indicates the area composed of dental pulp stem/stromal cells.
Comment 2
When using pulp slice model for cAMP assay how did the authors confirmed the recorded cells were actually odontoblast and not another cell type present in the pulp tissue?
Answer
Thank you for your valuable comments. As described above, we identified the cells as odontoblasts by their expression of DSPP. DSPP was highly expressed in the odontoblast layer and growing odontoblasts more than in the center area, which is composed of dental pulp stem/progenitor cells. It is important to leave the odontoblast single-cell level so that cAMP can be appropriately recorded after disconnecting the gap junction. We ensured that cells retained the high protein expression level of DSPP, which is known as a mature odontoblast marker, and that cells maintained their expression of DSPP and its function even when cells were growing out.
Comment 3
The figure 1 shows high magnification images of co-staining of DSPP with CALCRL/Gnas in dental pulp slices. It looks that odontoblasts are spread over a large area and do not form a single layer of cells as expected. And illustration at low magnification, showing all the pulp tissue, to confirm the location of the DSPP staining could help.
Answer
Thank you for pointing this out to us. Please see above this letter, which is the data on the immunocytochemistry of DSPP in dental pulp slices, including odontoblasts. DSPP was highly expressed in the odontoblast layer and growing odontoblasts more than in the central area, which is composed of dental pulp stem/progenitor cells. The odontoblasts were spread out, maintaining the cell layer.
Comment 4
The discussion and conclusion put this study in the context of pulp inflammation. Pulp inflammation generally happens when the tooth is developed, but the present study used pups rats tissue in which the tooth is likely not fully developed. As biological mechanisms can be different during the development and in adults, do the authors assume the mechanism they describe applies to adults teeth? Some references supporting that pups rat teeth are physiologically similar to adult rat teeth could be helpful to support the conclusion.
Answer
Thank you for your valuable suggestion. To model inflammatory responses in dental pulp, we established odontoblast-neuron co-culture. Isolation from adult teeth is very difficult, and the efficiency of collecting cells and the viability of cells are quite low, however. In addition, we conducted all experiments with incisors in the present study. Rodent incisors have dental stem cells in the dental mesenchyme and the epithelium in the incisor apical region. This unique anatomical feature is observed throughout their life. Based on these points, we considered no difference between the age groups.
Reviewer 3 Report
In the article: “Gαs-coupled CGRP receptor signaling axis from the trigeminal ganglion neuron to odontoblast negatively regulates dentin mineralization” the authors discussed about the application of an adenylyl cyclase (AC) activator and calcitonin gene-related peptide (CGRP) receptor agonist increasing the intracellular cAMP levels ([cAMP]i) in odontoblasts. Thus CGRP and cAMP signaling negatively regulate dentinogenesis as defensive mechanisms.
Overall, this manuscript results very interesting, the authors clearly explain the rational of the study and discussed the topic point by point.
However, we would like to invite the authors to clarify some minor points:
1. Please check the check punctuation and spaces;
2. Among the materials and methods section, the authors described the protocol for TG cells isolation, but they did check the phenotype by the expression of specific biomarkers?
3. Concerning the isolation of TG cells, there is a reference or it is a new protocol? Please specify among the text;
4. Why the TG cells were isolated from so young animal? (neonatal);
5. Which passage of TG cells in vitro culture was used for the experiments? Please specify among the text;
6. Figure 1: what is the used magnification?
7. Figure 5: the same of Figure 1;
8. Figure 6: the same of Figure 1 and 5.
Author Response
Dear Editor and Reviewer 3,
We deeply appreciate the constructive comments and suggestions of the editor and Reviewer 3. Thank you for reviewing our manuscript and offering valuable advice. We have addressed your comments with point-by-point responses and have revised the manuscript accordingly.
To Reviewer 3:
Comment 1
In the article: “Gαs-coupled CGRP receptor signaling axis from the trigeminal ganglion neuron to odontoblast negatively regulates dentin mineralization” the authors discussed about the application of an adenylyl cyclase (AC) activator and calcitonin gene-related peptide (CGRP) receptor agonist increasing the intracellular cAMP levels ([cAMP]i) in odontoblasts. Thus CGRP and cAMP signaling negatively regulate dentinogenesis as defensive mechanisms.
Overall, this manuscript results very interesting, the authors clearly explain the rational of the study and discussed the topic point by point. However, we would like to invite the authors to clarify some minor points: Please check the check punctuation and spaces;
Answer
Thank you for pointing this out. Accordingly, we have thoroughly checked the manuscript.
Comment 2
Among the materials and methods section, the authors described the protocol for TG cells isolation, but they did check the phenotype by the expression of specific biomarkers?
Answer
Thank you for pointing this out to us. Yes, we used a combination of cell size, the potency of depolarization-induced Ca2+ influx, and the expression pattern of CGRP and NF-H to identify small CGRP-positive and NF-H-negative TG neurons. To characterize the potential neurons for mechanical stimulation experiments, we prospectively distinguished the differences between TG neurons and other types of cells by inducing Ca2+ influx during the application of a solution containing a high concentration of K+ (50 mM K+) that promotes plasma membrane depolarization. In our study, neurons were retrospectively identified based on the expression patterns of NF-H and CGRP, as shown in Fig. 5. CGRP-positive cells were mainly negative for IB4, a non-peptidergic neuronal biomarker, as shown in this response letter.
Immunofluorescence staining of co-cultured cells revealed that CGRP was not expressed in odontoblasts.
The left photograph shows co-cultured cells composed of both odontoblasts and TG neurons. The image on the right is the magnified area outlined by the rectangle. CGRP-positive cells were negative for IB4, a non-peptidergic neuron biomarker. White arrowheads indicate odontoblasts and yellow arrowheads indicate TG neurons.
Please see the yellow highlight in the revised manuscript as shown below.
Page 10, line 345 to page 10, line 348
3.5. Direct mechanical stimulation of the TG neurons simultaneously increases the [Ca2+]i in the neuron and the [cAMP]i in the odontoblasts, approximating the stimulated neuron
TG cells were identified by the application of a solution containing a high concentration of K+ (50 mM), which induces membrane depolarization to specifically reveal neuronal intracellular Ca2+ responses in primary culture.
Comment 3
Concerning the isolation of TG cells, there is a reference or it is a new protocol? Please specify among the text;
Answer
We established an acute TG isolation. This original method has been reported in many articles over the past 10 years. In addition to cited references [12, 23-26], please refer to the following articles.
Kuroda et al., Mol Pain, 2013
Sato et al., Journal of Endodontics, 2018
Higashikawa et al., Front Cell Neurosci, 2019
Terashima et al., J Physiol Sci, 2019
Inoue et al., Int J Mol Sci, 2021
Comment 4
Why the TG cells were isolated from so young animal? (neonatal);
Answer
Thank you for your valuable comments. The calvaria protecting the brain is too hard to isolate TG neurons acutely in adult rats. Another reason is that connective tissues in adults make it difficult to isolate TG neurons appropriately. To focus on cellular interactions after mechanical stimulation, we used neonatal TG neurons. We have reported this method as described above.
Comment 5
Which passage of TG cells in vitro culture was used for the experiments? Please specify among the text;
Answer
Thank you for your valuable comments. The primary cultured cells were used in this study. This method is explained on page 3, line 127 to page 3, line 138.
Page 3, line 127 to page 3, line 138
Comment 6
Figure 1: what is the used magnification?
Comment 7
Figure 5: the same of Figure 1;
Comment 8
Figure 6: the same of Figure 1 and 5.
Answer
Thank you for pointing this out to us. We inserted scale bars for each photograph instead of inserting objective lens magnification information. The information in the revised manuscript is as follows.
p.6 3.1 Expression of heterotrimeric Gnas and CALCRL in rat odontoblasts
Data were obtained using a 40X magnification objective lens under a microscope. Enlarged images are shown as representative areas.
Answer
p.12
3.5. Direct mechanical stimulation of the TG neurons simultaneously increases the [Ca2+]i in the neuron and the [cAMP]i in the odontoblasts, approximating the stimulated neuron
Data were obtained using a 120X magnification objective lens under a microscope. Enlarged images are shown as representative areas.
Answer
p.13
3.6. CGRP-CGRP receptor signaling regulates defensive dentin demineralization
Data were obtained using a 20X magnification objective lens under a microscope. Each dataset was tiled with 63 photos to acquire the constructed data. To obtain data on the mineralized area constituted by odontoblasts, the mineralized area in the total area (I) was divided by the mineralized area of the dental pulp (I0).

Round 2
Reviewer 1 Report
The authors mostly asked the reviewer questions. This is satisfactory. However, there are little suggestions/comments as follows:
1). For anatomy, the matured odontoblasts are situated between the pre-dentin and dental pulp. In general, the matured odontoblasts are a layer. The odontoblasts extend their odontoblast processes into the pre-dentin and dentin and contribute the nutrition to these tissues. In the Materials and Methods, 2.3. Dental pulp slice preparation section, “The mandibular sections were sliced to ensure direct visibility of the dentin and enamel between the bone tissue and dental pulps. The surrounding impression material, bone tissue, enamel, and dentin were carefully removed and the remaining dental pulp slices were obtained” When the dentin is removed, the odontoblast processes would be separated from the odontoblasts. The matured odontoblasts will not be survivable. Therefore, the “odontoblasts” in the manuscript may be “dental pulp mesenchymal cells”. So far, based on the reviewer knowledge, it is hard to isolate “primary odontoblasts”. In this manuscript, the ‘odontoblasts” may be used “odontoblasts/dental pulp mesenchymal cells”.
2). In lines, 437-439, “In this study, immunostaining of co-cultured cells composed of TG cells and odontoblasts revealed that endogenous CGRP expression was not activated in odontoblasts by TG co-culture (personal communication with TO)”. In the sentence, it is confusing. In Figure 5, mechanical stimulation of a single TG neuron, but not the odontoblasts, in the co-culture (TN-odontoblasts) stimulated cAMP levels in the odontoblasts, it may indicate that the stimulated NTs release CGRP, which binds to its receptor in the odontoblasts and activate cAMP via AC. However, injury and inflammation are able to induce endogenous CGRP activity in the stimulated odontoblast/dental pulp mesenchymal cells. So, the sentence may be rewritten or re-adjusted.
Author Response
Dear Editor and Reviewer 1,
We deeply appreciate the constructive and productive comments and suggestions of the editor and reviewer. Thank you for reviewing our manuscript and offering valuable advice. We have addressed your comments in a point-by-point format and have revised the manuscript throughout.
Reviewer 1:
Comment 1
The authors mostly asked the reviewer questions. This is satisfactory. However, there are little suggestions/comments as follows:
1). For anatomy, the matured odontoblasts are situated between the pre-dentin and dental pulp. In general, the matured odontoblasts are a layer. The odontoblasts extend their odontoblast processes into the pre-dentin and dentin and contribute the nutrition to these tissues. In the Materials and Methods, 2.3. Dental pulp slice preparation section, “The mandibular sections were sliced to ensure direct visibility of the dentin and enamel between the bone tissue and dental pulps. The surrounding impression material, bone tissue, enamel, and dentin were carefully removed and the remaining dental pulp slices were obtained” When the dentin is removed, the odontoblast processes would be separated from the odontoblasts. The matured odontoblasts will not be survivable. Therefore, the “odontoblasts” in the manuscript may be “dental pulp mesenchymal cells”. So far, based on the reviewer knowledge, it is hard to isolate “primary odontoblasts”. In this manuscript, the ‘odontoblasts” may be used “odontoblasts/dental pulp mesenchymal cells”.
Answer
Thank you for your comments. The authors respectfully disagree that mature odontoblasts will not survive dental pulp slice preparation. Although the odontoblast processes are separated from their cell bodies during slice preparation, the cells are viable not only immediately after isolation but also after primary culture (e.g., PMIDs 12510803, 19828889, 22656960, 29709295, 24358160, 20307742, 27084672, 24939701, 21197505, and 29311993). This is because the dentin removed from the mandibular slices is thin and the processes are relatively short. In addition, the primary cultured odontoblasts of the dental pulp slices, which are positive for odontoblast marker proteins DSPP/DMP1/Nestin, allow us to conduct research using single living odontoblasts, that is, patch-clamp recording, intracellular free calcium ion, and/or cAMP concentration measurements. This indicates that the isolated odontoblasts are viable. The success rate of whole-cell patch-clamp recordings, for example, are dramatically reduced by reducing cell viability. The success rate of whole-cell patch-clamp recordings using single live primary cultured odontoblasts obtained by dental pulp slice preparation is high.
In addition, as the reviewer states, it is difficult to isolate living odontoblasts because of their anatomical location, and the survival rate of cells obtained by scratching dentin is relatively low. To solve this problem, we developed a dental pulp slice preparation method for isolating “healthy” odontoblasts, inspired by brain slice preparation; we successfully obtained odontoblasts for this experiment at the single living cell level. We hope that the reviewer understands this point.
In response to this comment, however, we have added a sentence to the Materials and Methods section, as follows: Page 3 line 119 to page 3 line 120; “We selected mandible sections with thin dentin (but with enamel and dentin distinguishable under a microscope) to avoid cellular damage in odontoblasts.”
In addition, we have attached the Figure below showing dental pulp slice preparation published in Cell Calcium (2012).
Comment 2
2). In lines, 437-439, “In this study, immunostaining of co-cultured cells composed of TG cells and odontoblasts revealed that endogenous CGRP expression was not activated in odontoblasts by TG co-culture (personal communication with TO)”. In the sentence, it is confusing. In Figure 5, mechanical stimulation of a single TG neuron, but not the odontoblasts, in the co-culture (TN-odontoblasts) stimulated cAMP levels in the odontoblasts, it may indicate that the stimulated NTs release CGRP, which binds to its receptor in the odontoblasts and activate cAMP via AC. However, injury and inflammation are able to induce endogenous CGRP activity in the stimulated odontoblast/dental pulp mesenchymal cells. So, the sentence may be rewritten or re-adjusted.
Answer
Thank you for pointing this out. As the reviewer suggests, some studies revealed that during injury and inflammation, such as pulpitis, dental pulp cells increasingly express endogenous CGRP. In the first revision round from reviewer 1, we have learned that Fehrenbacher et al., reported the expression pattern of CGRP in nerve fibers located in the human dental pulp (Fehrenbacher et al., Pain, 2009). And we also have learned that Caviedes-Bucheli et al., reported that CGRP receptor expression in human pulp tissue is significantly increased during inflammatory phenomena such as acute irreversible pulpitis (Caviedes-Bucheli et al., Int Endod J, 2005). In the present study, we did not observe any immunoreactivity of endogenous CGRP protein expression level in “odontoblasts”.
We totally agree further studies are needed to clarify whether odontoblasts endogenously express CGRP specifically during pulpal inflammation. Additionally, the mechanism of CGRP signaling during pulpal inflammation is of interest. We have thus modified the following sentence in the Discussion section and added two articles to the references to clarify this point: Page 14, line 440 to page 14, line 446, “Nerve fibers in dental pulp express CGRP [33] and dental pulp cells express CGRP receptors increasingly during inflammatory phenomena such as acute irreversible pulpitis [34]. In this study, endogenous CGRP expression could not be observed in physiological conditions of odontoblasts using immunostaining (personal communication with TO), further study will be needed to reveal whether the inflammatory response induces endogenous CGRP expression in odontoblasts.”
Reviewer 2 Report
I thanks the Authors to have addressed all my comments.
Author Response
Thank you for reviewing our manuscript.